# Mapping and Identifying Candidate Genes Enabling Cadmium Accumulation in *Brassica napus* Revealed by Combined BSA-Seq and RNA-Seq Analysis

**DOI:** 10.3390/ijms241210163

**Published:** 2023-06-15

**Authors:** Huadong Wang, Jiajia Liu, Juan Huang, Qing Xiao, Alice Hayward, Fuyan Li, Yingying Gong, Qian Liu, Miao Ma, Donghui Fu, Meili Xiao

**Affiliations:** 1Key Laboratory of Crop Physiology, Ecology and Genetic Breeding, Ministry of Education, Agronomy College, Jiangxi Agricultural University, Nanchang 330045, China; wanghuadong_jxau@163.com (H.W.); 15838199651@163.com (J.L.); 15907073169@163.com (J.H.); lifuyann1@163.com (F.L.); kutn3139@163.com (Y.G.); 15211092301@163.com (Q.L.); mm15308249047@163.com (M.M.); fudhui@163.com (D.F.); 2Graduate School of Jiangxi Normal University, Jiangxi Normal University, Nanchang 330045, China; xiaoqing_04@163.com; 3Centre for Horticultural Science, Queensland Alliance for Agriculture and Food Innovation, The University of Queensland, Brisbane 4072, Australia; a.hayward@uq.edu.au

**Keywords:** cadmium accumulation, *Brassica napus*, bulk segregation analysis, whole genome resequencing, transcriptome

## Abstract

Rapeseed has the ability to absorb cadmium in the roots and transfer it to aboveground organs, making it a potential species for remediating soil cadmium (Cd) pollution. However, the genetic and molecular mechanisms underlying this phenomenon in rapeseed are still unclear. In this study, a ‘cadmium-enriched’ parent, ‘P1’, with high cadmium transport and accumulation in the shoot (cadmium root: shoot transfer ratio of 153.75%), and a low-cadmium-accumulation parent, ‘P2’, (with a cadmium transfer ratio of 48.72%) were assessed for Cd concentration using inductively coupled plasma mass spectrometry (ICP-MS). An F_2_ genetic population was constructed by crossing ‘P1’ with ‘P2’ to map QTL intervals and underlying genes associated with cadmium enrichment. Fifty extremely cadmium-enriched F_2_ individuals and fifty extremely low-accumulation F_2_ individuals were selected based on cadmium content and cadmium transfer ratio and used for bulk segregant analysis (BSA) in combination with whole genome resequencing. This generated a total of 3,660,999 SNPs and 787,034 InDels between these two segregated phenotypic groups. Based on the delta SNP index (the difference in SNP frequency between the two bulked pools), nine candidate Quantitative trait loci (QTLs) from five chromosomes were identified, and four intervals were validated. RNA sequencing of ‘P1’ and ‘P2’ in response to cadmium was also performed and identified 3502 differentially expressed genes (DEGs) between ‘P1’ and ‘P2’ under Cd treatment. Finally, 32 candidate DEGs were identified within 9 significant mapping intervals, including genes encoding a glutathione S-transferase (GST), a molecular chaperone (DnaJ), and a phosphoglycerate kinase (PGK), among others. These genes are strong candidates for playing an active role in helping rapeseed cope with cadmium stress. Therefore, this study not only sheds new light on the molecular mechanisms of Cd accumulation in rapeseed but could also be useful for rapeseed breeding programs targeting this trait.

## 1. Introduction

The expansion of industrial activity and the use of cadmium-containing chemical fertilizers and products have led to increased cadmium pollution in farmlands, which poses significant harm to plants, the environment, and human health [1,2,3]. Phytoremediation, a low-cost, sustainable, and environmentally friendly method that uses hyperaccumulator plants to selectively remove and recover heavy metals, was first proposed by Chaney in 1983 [4,5]. Globally, researchers have identified approximately 450 plant species that are capable of accumulating heavy metals, including 277 species that are particularly rich in nickel. Examples of such nickel-accumulated plants include *Cheiranthus* and *Thlaspi arvense* in the *Cruciferous* family, *Chinese leaf subflora* in the *Euphorbiaceae* family, and *Senecio* in the *Asteraceae* family [6,7,8,9]. Examples of copper-rich plants include those found in the *Labiatae* family [10], and cobalt-rich plants can be found in the *Scrophulariaceae*, *Myrtaceae,* and *Brassicaceae* families amongst others [11,12,13]. Other plants rich in heavy metals, such as arsenic, zinc, cadmium, selenium, and manganese, have also been identified [14,15,16,17].

Rapeseed (*Brassica napus* L.), an edible oil crop, is globally recognized for its large biomass, ease of harvesting, and strong climate adaptability. However, it has been reported that *Brassica* crops accumulate high concentrations of toxic metals, including Cd and Pb, as well as other metals such as Cr, Cu, Ni, and Zn [18,19]. Different *Brassica* species have different tolerance to Cd and vary in the organs that accumulate Cd. The highest distribution of cadmium in rapeseed organs has been found in leaves, followed by stems, roots, and pods, with the lowest accumulation in seeds [20,21]. Therefore, rapeseed remediation of cadmium-contaminated soil could prove an efficient, economical, and environmentally friendly approach, as the oil extracted from rapeseed grown in polluted areas could still be used in power plants or biodiesel, promoting a circular economy [19,22,23].

Cadmium often inhibits plant growth, and plants have a complex and interconnected network of defense strategies against cadmium to avoid or tolerate heavy metal poisoning. These strategies can generally be categorized into three main types: inhibition of cadmium uptake, inhibition of cadmium transport, and sequestration of cadmium in vacuole compartments [16,24]. The accumulation of cadmium begins with passive and active transport of cadmium ions from the soil through roots. Heavy metal ATPases (HMAs) are involved in the intracellular and intercellular transport of cadmium, with the genes *HMA1*, *HMA2*, *HMA3*, and *HMA4* playing crucial roles in the translocation and segregation of cadmium from the roots to the shoot, as shown in rice, wheat, and other species [21,25,26,27,28].

Different plant species behave somewhat differently under cadmium stress; for example, the Cd uptake gene *Cdu1* controls cadmium concentration in durum wheat grain by limiting uptake, whereas in soybean, a different gene, *Cda1*, is responsible for the uptake of cadmium from the soil through the roots of the plant [29,30]. In rice, the ATP binding cassette (ABC)-type transporter gene *OsABCG43* is described as a Cd importer that controls Cd accumulation in the vasculature of leaves and roots, while *OsNramp5* loss-of-function mutants show significantly reduced cadmium concentrations in roots, shoots, and grains [31].

Rapeseed has a strong ability to accumulate cadmium. However, few studies have focused on the study of Cd stress in *Brassica napus*. For this reason, here, low-cadmium-accumulation and cadmium-enriched (high accumulation) rapeseed were identified. An F_2_ population was constructed, and, using combined bulk segregant analysis with high-throughput resequencing, cadmium-enriched QTLs were identified. Transcriptomic analysis revealed several differentially expressed candidate genes in mapped QTL regions. These are presented as key genes associated with Cd response and accumulation in *B. napus*. This study highlights the molecular mechanisms of cadmium enrichment in rapeseed and provides a theoretical basis for the remediation of cadmium-contaminated soil using rapeseed.

## 2. Results

### 2.1. Morphological Observation under Cadmium Stress

Two rapeseed lines ‘P1’ (cadmium-enriched line) and ‘P2’ (low-cadmium-accumulation line) were selected under 5 mg/kg cadmium treatment. The ‘P2’ rapeseed line exhibited delayed growth, leaves changing from green to yellow, and leaf senescence, while the cadmium-enriched line ‘P1’ did not show any obvious morphological changes (Figure 1A,B). There was a significant decrease in the chlorophyll-a content of the ‘P2’ parent under cadmium treatment, while the chlorophyll-b content in both ‘P1’ and ‘P2’ did not significantly decrease after being subjected to Cd treatment (Figure 1C,D). This resulted in a significant decrease in total chlorophyll content in ‘P2’ after Cd treatment (Figure 1E). ICP-MS was employed to determine the ionomic basis of the differential Cd accumulation between ‘P1’ and ‘P2’. The shoot, root, and whole plant Cd concentrations were significantly higher in ‘P1’ than in ‘P2’ (Figure 1F–H). Furthermore, Cd concentration in the shoot was significantly higher than in the roots in the ‘P1’ line, and the transfer ratio (translocation of Cd from the roots to the shoot of a plant) was up to 153.75% in the ‘P1’ line versus 48.72% in the ‘P2’ line (Figure 1I).

### 2.2. Construction and Genetic Analysis of a Segregating Population for Cadmium-Enrichment

To analyze the underlying genetics of cadmium enrichment traits in rapeseed, an F_2_ population was generated by crossing the cadmium-enriched line ‘P1’ with the low-cadmium-accumulation line ‘P2’. The growth of F_2_ plants treated with 5 mg/kg CdCl_2_ was assessed and classified into seven categories: dead seedling, stagnant growth, slow growth, leaf bleaching, leaf senescence, dwarfing, and normal growth (Figure 2A). The accumulation of Cd^2+^ in the shoots and roots of the F_2_ population was analyzed (Appendix A), revealing a normal distribution of plant Cd^2+^ concentrations in the F_2_ segregating population (Figure 2B–D). This indicated that cadmium enrichment in *Brassica napus* is a quantitative trait controlled by multiple genes.

### 2.3. Identification and Verification of Cadmium-Enrichment-Related QTLs

To detect genomic variations underlying cadmium enrichment, an integrated bulked segregant analysis (BSA) and resequencing approach was used; BSA-seq. Specifically, both a low-cadmium pool and a cadmium-enriched pool were re-sequenced, consisting of DNA from 30 low-cadmium and 30 cadmium-enriched individuals from the F_2_ population, respectively. The total amount of clean data generated was 182.16 GB, with an average Q30 ratio of 91.10% and an average GC content of 37.21% (Appendix A). After filtering, a total of 3,660,999 SNPs and 787,034 InDels were identified by comparing the sequencing data to the reference genome ‘ZS11’. Based on the delta SNP index (the difference in SNP frequency between the two bulked pools), nine candidate intervals from five chromosomes were identified: ChrA09 18.3–19.1 Mb, ChrA09 35.1–35.6 Mb, ChrC02 32.35–32.85 Mb, ChrC02 34.95–35.45 Mb, ChrC02 44.45–44.95 Mb, ChrC03 30.25–30.75 Mb, ChrC02 37.05–37.55 Mb, ChrC05 0.60–1.85 Mb, and ChrC08 47.90–48.70 Mb (Figure 3, Table 1).

### 2.4. Verification of Cadmium-Enrichment-Related QTLs

To further validate QTL loci, KASP (Kompetitive Allele-Specific PCR) primers were developed based on resequencing data in the four intervals of the A09 and C03 chromosomes. Initially, 44 primers were used to detect polymorphisms in the parental lines P1 and P2 and in the hybrid F_1_. Out of these, eight primers produced clear genotyping results in the parent and hybrid F_1_ generation (Appendix A). These 8 pairs of primers were further used to genotype 89 low-cadmium-accumulation individuals from the F_2_ population. The results for the QTL loci on the A09 chromosome (Cd-en-qtl1 and Cd-en-qtl2) are shown in Figure 4A,B, while the results for the QTL loci on C03 (Cd-en-qtl6 and Cd-en-qtl7) are shown in Figure 4C,D. This revealed that the KASP primers developed based on these four QTL intervals can reliably identify low-cadmium-accumulation individuals from the F_2_ population and that the QTL loci identified in this study are reliable.

### 2.5. Identification of Differentially Expressed Genes between P1 and P2

High-throughput genome-wide transcriptome sequencing was performed on ‘P1+’ (P1 grown under Cd treatment), ‘P2+’ (P2 grown under Cd treatment), ‘P1’ (P1 without Cd treatment), and ‘P2’ (P2 without Cd treatment), generating a total of 41.9 Gb of clean reads (Appendix A). A Pearson correlation coefficient heatmap and principal component analysis showed good reproducibility among biological replicates and high discrimination between the samples (Figure 5A). FPKM (Fragments Per Kilobase Million) values were used to evaluate gene expression levels in the four samples.

A total of 83,713 genes were expressed, and Venn diagram analysis showed that 3502 DEGs (|log2FoldChange| ≥ 1, Q-value ≤ 0.05) were specifically responsive to Cd stress in ‘P1+’ compared to ‘P1’, ‘P2’, and ‘P2+’ (Figure 5B and Appendix A). Several genes were randomly selected for verification with quantitative reverse transcription polymerase chain reaction (qRT-PCR) (Appendix A). To further explore the functions of DEGs, we conducted a Gene Ontology (GO) and Kyoto Encyclopaedia of Genes and Genomes (KEGG) pathway enrichment (Q-value ≤ 0.05) analysis. GO enrichment analysis revealed that DEGs were significantly enriched in biological processes such as the carboxylic acid metabolic process (GO:0019752), oxoacid metabolic process (GO:0043436), organic acid metabolic process (GO:0006082), and metal ion transport (GO:0030001) (Figure 5C and Appendix A). Meanwhile, KEGG pathway enrichment analysis showed that 3502 DEGs participated in 217 pathways (Figure 5D and Appendix A). Of these, 25 pathways were significantly enriched including mitogen-activated protein kinase (MAPK) signaling, nitrogen metabolism, cysteine and methionine metabolism, linoleic acid metabolism, glucosinolate biosynthesis, pyruvate metabolism, and biosynthesis of other types of O-glycan.

### 2.6. Combined Analysis with BSA-Seq and RNA-Seq

To further identify candidate genes responsible for cadmium accumulation in ‘P1’, the results of the BSA-seq were integrated with RNA-seq. A total of 308 genes containing SNPs or InDels were identified in the BSA intervals, with 218 also found to be expressed genes in the RNA-seq. Among these, 32 differentially expressed genes exceeded the threshold value (genes with a difference multiplicity >2 and Q-value less than 0.05) and were identified as candidate genes related to the control of cadmium enrichment in ‘P1’. Of these, 10 genes were up-regulated in the low-Cd-accumulation ‘P2’ line, and 22 genes were up-regulated in the Cd-enriched ‘P1’ line. Moreover, 26 genes contained SNPs or InDels in the coding region, 1 gene contained SNPs in the 5′ regulatory region (2000-bp upstream), and 5 genes contained SNPs or InDels in the intergenic regions. The functional information of these 32 genes was annotated (Table 2). Prime candidates included BnaA09G0263300ZS—a glutathione S-transferase that plays a major role in Cd stress response [32,33], BnaC02G0320100ZS—a phosphoglycerate kinase (PGKc) important for the maintenance of cellular pH homeostasis [34], BnaC03G0450600ZS—a DnaJ (Hsp40) protein involved in maintaining the correct folding, assembly, transport, and degradation of intracellular proteins and that plays an important role in the resistance and adaptation of organisms to high and low temperatures, drought, and other stresses, and BnaC08G0448600ZS—orthologous to stress-enhanced protein 2 (SEP2), which contains two transmembrane segments and has been reported to be involved in the response of plants to abiotic stress [35].

## 3. Discussion

Rapeseed is one of the most economically important oil crops in the world. Adding to this is its strong capacity for cadmium absorption and accumulation combined with high biomass, making it attractive for cadmium restoration relative to other reported hyperaccumulator plants [19,36,37]. In this study, mature plants of a cadmium-enriched line (P1) treated with 5 mg/kg cadmium achieved a total cadmium content of 117.29 mg/kg, with a cadmium root-to-shoot transfer ratio reaching 153.75%. This indicates that rapeseed has a strong potential for soil improvement under heavy cadmium metal pollution.

Cadmium contamination negatively affects the photosynthetic system in plants. It can cause swelling and distortion of chloroplasts and disintegration of the chloroplast vesicles of leaf cells. Studies have shown that in Cd-hyperaccumulator plants, Cd is often transported to vacuoles, particularly the lytic vacuole located in the leaf. This process helps to minimize the damaging effects on the leaf caused by exposure to Cd [38,39,40]. Conversely, in low-Cd-accumulation cultivars, Cd remains in the cytoplasm and plastids, preventing electron transport in the stroma/thylakoid and ultimately leading to carbon deficiency [38,39,40]. Since the green color of leaves is due to the presence of chlorophyll, plant leaves under cadmium stress will appear yellowish, yellow, or whitish-yellow. In severe cases, the electron transport and photosynthetic capacity of chloroplasts are inhibited, resulting in plant death [41,42,43,44].

In the present study, the total chlorophyll content of the cadmium-enriched ‘P1’ line did not decrease significantly, while the total chlorophyll content of low-cadmium-accumulation line ‘P2’ decreased significantly under cadmium treatment, accompanied by yellowing and ageing of the leaves. This indicates that the photosynthesis of the plants was affected, ultimately inhibiting their growth and development.

Several studies have analyzed the cadmium response mechanism in rapeseed through comparative analysis of the transcriptome and proteome [42,45,46]. However, few studies have reported QTLs or genes related to cadmium accumulation in rapeseed from a population genetics perspective. The QTL mapping approach is highly important for innovating marker-assisted selection for crops [47], enabling rapid breeding for traits of value. Therefore, in this study, a genetic population segregated for cadmium hyperaccumulation was constructed and investigated using an innovative approach combining BSA analysis with resequencing and RNA-seq technology to obtain QTLs and strong candidate genes associated with cadmium accumulation in rapeseed. Nine candidate intervals were identified from five chromosomes: ChrA09 18.3–19.1 Mb, ChrA09 35.1–35.6 Mb, ChrC02 32.35–32.85 Mb, ChrC02 34.95–35.45 Mb, ChrC02 44.45–44.95 Mb, ChrC03 30.25–30.75 Mb, ChrC03 37.05–37.55 Mb, ChrC05 0.60–1.85 Mb, and ChrC08 47.90–48.70 Mb. Among these candidate intervals, chromosomal loci such as ChrA09 18.3–19.1 Mb, ChrC02 44.45–44.95 Mb, and ChrC08 47.90–48.70 Mb have also been identified in a previous genome-wide association study of rapeseed under 5 mg/kg cadmium stress [48]. These findings not only provide a good basis for understanding the genetic control of cadmium accumulation in rapeseed but will also enable further development of molecular markers to assist the breeding of high cadmium-accumulating rapeseed. Cadmium is an essential component of nickel–cadmium (“Ni-Cad”) batteries, used in multiple applications from household appliances to emergency power supplies in hospitals. Ultimately, the development of hyperaccumulating selections could enable simultaneous cultivation and remediation of cadmium-polluted soils, ensuring both economic (oil and cadmium harvest) and environmental outcomes with minimal biomass waste.

Plants exposed to toxic metals experience significant stress, as these metals have the ability to convert hydrogen peroxide into hydroxyl and hydroperoxyl radicals, which are major forms of reactive oxygen species (ROS). ROS produced in response to cadmium (Cd) exposure have been shown to damage plant membranes and cause destruction of cell biomolecules and organelles [49]. This can result in reduced plant uptake of essential minerals such as iron (Fe) and zinc (Zn) as well as lower rates of photosynthesis, ultimately leading to a decline in crop production and quality [50]. Excessive accumulation of ROS can lead to the destruction of biofilms, DNA backbone, functional proteins, and programmed cell death in plants [51,52]. In the present study, RNA-seq analysis was performed on young rapeseed seedlings exposed to Cd, resulting in the identification of 32 differentially expressed genes within 9 significant QTL candidate intervals. It is noted that if individual tissues (roots, stems, and leaves) were collected and sequenced separately, tissue-specific differentially expressed genes would likely be identified that may have been missed here. One of the candidate genes, *BnaA09G0263300ZS*, was found to contain five nonsynonymous SNPs and was upregulated after 5 mg/kg cadmium treatment in cadmium-enriched ‘P1’. This gene encodes a glutathione S-transferase (GST). GST has been previously reported to play a variety of roles in plant responses to Cd stress by binding exogenous toxic molecules to glutathione [[53,54,55]]. For example, in rice, expression of the *GST* genes *OsGSTU5* and *OsGSTU37* was induced in the presence of cadmium, and overexpression in both yeast and rice enhanced cadmium tolerance. *GST* has also been reported to be associated with cadmium tolerance in bacteria, animals, and humans [56,57].

Several additional candidate genes with potentially important biological functions for enhancing cadmium tolerance in rapeseed have been identified here. For example, the candidate gene *BnaC02G0320100ZS* encodes a phosphoglycerate kinase (PGK). Studies have shown that ectopic expression of phosphoglycerate kinase-2 (*OsPGK2-P*) in tobacco improves salt tolerance [34]. It has also been shown that *PGK1* and *PGK3* are involved in photosynthetic carbon metabolism to optimize growth by reducing photosynthetic activity when the glycolytic pathway is impaired [58].

Cadmium can exert its toxic effects by inducing misfolding of nascent proteins [59,60]. The DnaJ family is the most diverse family of co-chaperones that normally act to facilitate correct folding, and the conserved J domain is typical of its members. DnaJ is involved in different physiological roles by processing different substrates [61,62]. In this study, candidate genes *BnaC08G0467400ZS* and *BnaC03G0450600ZS* both encode a molecular chaperone DnaJ protein. *BnaC03G0450600ZS* was found to contain a nonsense SNP mutation and was downregulated after 5 mg/kg cadmium treatment in low-cadmium-accumulation ‘P2’. No SNP was found in the *BnaC08G0467400ZS* coding region, but the upregulated expression of *BnaC08G0467400ZS* seen in ‘P1’ may mitigate the toxic effects of increased misfolded proteins in rapeseed under Cd stress by increasing the folding rate of correctly folded proteins. *BnaC08G0448600ZS* encodes a stress-enhanced protein 2 (SEP2), which was upregulated in ‘P2’ and has been reported to play an important role in the normal functioning of chloroplasts under light stress [35]. Additional candidate genes upregulated specifically in P1 and with SNPs (nonsynonymous) in their coding regions were a LURP-one-related protein (BnaC*08G0466900ZS*), a 26S proteasome, *(BnaC03G0447700ZS*), and a receptor-like protein kinase (*BnaC03G0448400ZS*). These candidate genes may play indirect roles in helping plants cope with cadmium stress [63,64,65,66,67]. Furthermore, there were five unannotated protein-coding genes (*BnaC08G0457500ZS*, *BnaC08G0424900ZS*, *BnaC03G0450200ZS, BnaC03G0448500ZS*, and *BnaA09G0264000ZS*) whose biological mechanisms involved in cadmium tolerance in rapeseed need further exploration. It is possible that unknown genes could have functions involved in various diverse mechanisms of Cd tolerance and accumulation, such as the transport and sequestration of Cd in the lytic vacuoles to prevent leaf damage [68,69,70], and not only in mediating or detoxifying the stress response.

Transcription factors such as WRKY and MYB play important roles in regulating plant abiotic stress response [71,72,73,74]. For instance, in soybean, GmWRKY142 binds and activates the transcription of *ATCDT1*, *GmCDT1-1*, and *GmCDT1-2* and significantly enhances Cd tolerance [75]. *ZmWRKY4* plays an essential role in the upregulation of the expression and activity of antioxidant enzymes, such as superoxide dismutase (SOD) and ascorbate peroxidase (APX), under Cd stress [76]. Similarly, AtMYB4 enhances the antioxidant activity of *Arabidopsis thaliana* by activating the transcription of *phytochelatin synthase 1 (PCS1)* and *metallothionein 1C (MT1C*), thereby improving cadmium tolerance [77]. Meanwhile, overexpression of *AtMYB49* in Arabidopsis resulted in a significant increase in Cd accumulation whereas *myb49* knockout plants showed a decrease in Cd accumulation [78].

Among the candidate genes in this study, two genes (*BnaC08G0466100ZS* and *BnaC08G0467300ZS*) encoding transcription factor family proteins were identified. *BnaC08G0466100ZS* contained a synonymous SNP and showed significant upregulation in the cadmium-enriched ‘P1’ line while *BnaC08G0467300ZS* contained serval SNPs in the regulatory region, and showed significant downregulation in the cadmium-enriched ‘P1’ line. It is plausible that these two genes may regulate Cd stress response by controlling the expression of downstream genes and are central components of the Cd detoxification, vacuole sequestration, or tolerance regulatory network.

## 4. Materials and Methods

### 4.1. Plant Material and Growth Conditions

In this study, high cadmium-enriched rapeseed line ‘P1’ (18M08485) and low-cadmium-accumulation rapeseed line ‘P2’ (zheyou28) were identified from a total of 186 rapeseed germplasm resources using germination tests under different gradients of cadmium concentration (5 mg/kg, 10 mg/kg, 20 mg/kg, 40 mg/kg, and 60 mg/kg) treatment and using a 2-year contaminated soil potting test trial (data unpublished). High cadmium-enriched rapeseed line ‘P1’ and low cadmium accumulation rapeseed line ‘P2’ were selected to generate an F_2_ population. The soil used in the study was sieved to remove debris larger than 1 cm in diameter. Considering that an F_2_ genetic population was planted to investigate the genetic basis of cadmium accumulation of *B. napus* in this experiment, a relatively low soil cadmium concentration was employed; the Cd^2+^ content was adjusted to 5 mg/kg by adding CdCl_2_·2.5H_2_O. All the plants were in pots in greenhouses located at Jiangxi Agricultural University in autumn (115.8 E, 28.8 N).

### 4.2. Pigment Content Determination

‘P1’ and ‘P2’ plants were grown in soil containing 5 mg/kg CdCl_2_·2.5H_2_O (5 mg of cadmium per 1 kg of soil) for 45 days. Leaves weighing approximately 0.2 g fresh weight were cut into pieces and immersed in 10 mL of 80% (*v*/*v*) acetone for 48 h. The mixture was then centrifuged at 3000 r/min for 10 min, and the concentrations of total chlorophyll in ‘P1’ and ‘P2’ were determined using a spectrophotometer (UV-1800, Mapada, Shanghai, China) at 663 nm, 646 nm, and 470 nm, as described in [79]. Statistically significant difference was analyzed utilizing unpaired *t*-test.

### 4.3. Measuring the Concentration of Cd

To analyze Cd accumulation, the aboveground and underground parts of all mature F_2_ rapeseed plants were harvested and dried in a 65 °C oven for 3 days until reaching a constant weight. The samples were ground into powder using a hammer cyclone mill. For the aboveground samples, 0.2 g was weighed, and for the underground samples, 0.1 g was weighed. Then, 8 mL of nitric acid and 2 mL of 30% hydrogen peroxide solution were added and the samples were digested in a microwave digester with the following digestion procedure: 150 °C for 10 min, and 180 °C for 20 min. The sample was then diluted to 50 mL with double-distilled water, and the cadmium concentration was determined using inductively coupled plasma mass spectrometry (ICP-MS; NexIONTM 350×, PerkinElmer, The United States of America) [80]. The analysis process of the sample Cd followed the instructions of the national standard GB 5009.268-2016, issued by National Standard Substances Center of China. Statistically significant differences were analyzed using a paired *t*-test.

### 4.4. DNA Isolation and Whole-Genome Re-Sequencing (WGS)

In this study, 50 cadmium-enriched F_2_ individuals and 50 low-cadmium-accumulation F_2_ individuals were screened based on cadmium content and cadmium transfer ratio. DNA was extracted from 100 rapeseed leaf samples using the CTAB method with modifications [81]. The extracted DNA concentration was adjusted to 100 ng/μL. DNA from cadmium-enriched F_2_ individuals was mixed to create an F-bulk pool, while DNA from low-accumulation rapeseed F_2_ individuals was mixed to create a D-bulk pool. The cadmium-enriched parents formed the P1 pool, and the low-cadmium-accumulation parents formed the P2 pool. Whole-genome resequencing was performed using Illumina’s HiSeq2000 platform (BGItech, Shenzhen, China) at a depth of 50× per pooled sample. The reads were aligned to the ZS11.v0 reference genome [82] using Burrows-Wheeler Aligner (BWA) software bwa-0.7.10 [83], GATK toolkit used to detect and filter SNPs [84], and candidate intervals were obtained using SNP index analysis with a window size of 0.5 Mb as described in [85].

### 4.5. Development of KASP Markers

To validate the polymorphisms using KASP^TM^ markers synthesized at Sangon Biotech (Shanghai, China), the parent lines and F_1_ hybrid were initially tested. KASP reactions were performed in 10 µL volumes, consisting of 5 µL of 25 ng/µL DNA template, 5 µL of 2× KASP master mixture, and 0.14 µL of primer assay mixture. PCR cycling was carried out as follows: an initial denaturation step at 94 °C for 15 min, followed by 10 touchdown cycles (94 °C for 20 s; touchdown at 61 °C initially and decreasing by 0.6 °C per cycle for 60 s), followed by 26 additional cycles of annealing (94 °C for 20 s; 55 °C for 60 s). Fluorescence was read using a Pherastar and analyzed using BMGPHERAstar Software.

### 4.6. RNA Sequencing and Transcriptomic Analysis

P1 and P2 seedlings were grown for three weeks in the presence (P1+ and P2+) or absence (P1 and P2) of 5 mg/kg cadmium. Total RNA was extracted from whole rapeseed seedlings (with three biological replicates) of each genotype and treatment group (P1, P2, P1+, and P2+) using the TaKaRa MiniBEST Plant RNA Extraction Kit (TaKaRa, Shiga, Japan) following the manufacturer’s instructions, with genomic DNA removed using TaKaRa DNase I. All RNA samples were stored at −80 °C. RNA concentration was measured three times using an Agilent 2100 (Agilent Technologies Inc., Santa Clara, CA, USA). Three independent libraries for each genotype and treatment were prepared using the TruSeq Stranded kit (Illumina, San Diego, CA, USA) following the manufacturer’s instructions. Sequencing (MiSeq Reagent Kit v3, 150 cycles) was performed on an Illumina MiSeq sequencer. Reads were mapped to the *Brassica napus* ‘ZS11’ genome (derived from accession SRS4884914) using HISAT2 [86], and the count data were analyzed using DESeq2 [87]. GO and KEGG enrichment analyses were performed using TBtools [88].

### 4.7. qRT-PCR Analysis

Several genes were verified using quantitative reverse transcription polymerase chain reaction (qRT-PCR) (Appendix A). Primers were designed using the Primer 3 website (https://bioinfo.ut.ee/primer3-0.4.0/ (accessed on 1 December 2021)). One microgram of RNA per sample was used to synthesize cDNA using Superscript II Reverse Transcriptase (Cat. number 18064-014, Invitrogen, Thermo Scientific, Thousand Oaks, CA, USA) and oligo-dT (12–18) (Invitrogen, Thermo Scientific, Thousand Oaks, CA, USA). qRT-PCR was conducted using the Bio-Rad CFX96 real-time system (Bio-Rad Laboratories, Hercules, CA, USA). Relative expression levels were calculated using the 2^−ΔΔCt^ method [89] with *BnaACTIN* used as the normalizer for total RNA level normalization in the qRT-PCR assays.

### 4.8. Statistical Analysis

Statistical parameters are shown in the figures and figure legends. qRT-PCR was generally performed in triplicate, and bars represent means ± SD. Statistically significant differences were determined using paired *t*-test and unpaired *t*-test: * *p* < 0.05, ** *p* < 0.01, and *** *p* < 0.001. Statistical analyses were performed using Microsoft Excel 2016. Image analyses were generated using R/ggplot2.

## 5. Conclusions

Hyperaccumulating plant species are of high biological and evolutionary interest as well as presenting exciting prospects for use in phytomining and phytoremediation. Rapeseed, known for its adaptability and large biomass, can efficiently remove cadmium from contaminated soils while retaining its economic value as an oil seed. In this study, we combined quantitative genetics with bulk segregant genome sequencing and transcriptome sequencing to identify candidate genes for cadmium hyperaccumulation in rapeseed. Nine significant QTL candidate intervals and thirty-two differentially expressed candidate genes in an F_2_ genetic population constructed from a Cd-enriched ‘P1’ line and low-Cd-accumulation ‘P2’ line were found. Of the resulting 27 annotated genes, *BnaA09G0263300ZS* encodes a glutathione S-transferase likely to enable Cd-enriched rapeseed to reduce the toxic effects of cadmium-induced reactive oxygen species. Other candidate genes, such as *BnaC03G0450600ZS* and *BnaC08G0467400ZS*, encode the molecular chaperone DnaJ, which could mitigate Cd toxicity by promoting correctly folded proteins under Cd stress. Other candidate genes, such as *BnaC08G0466100ZS* and *BnaC08G0467300ZS*, encode transcription factor family proteins that may be vital to responding to Cd stress by regulating the transcript levels of Cd-responsive genes. Additionally, five unannotated protein-coding genes were identified and will be further studied as potential cadmium accumulation genes. Therefore, the candidate intervals and genes identified in this study will lay the foundation for future elucidation of the regulatory mechanism of Cd accumulation in rapeseed and provide a basis for breeding for effective cadmium remediation in agricultural soils.

## Figures and Tables

**Figure 1 ijms-24-10163-f001:**
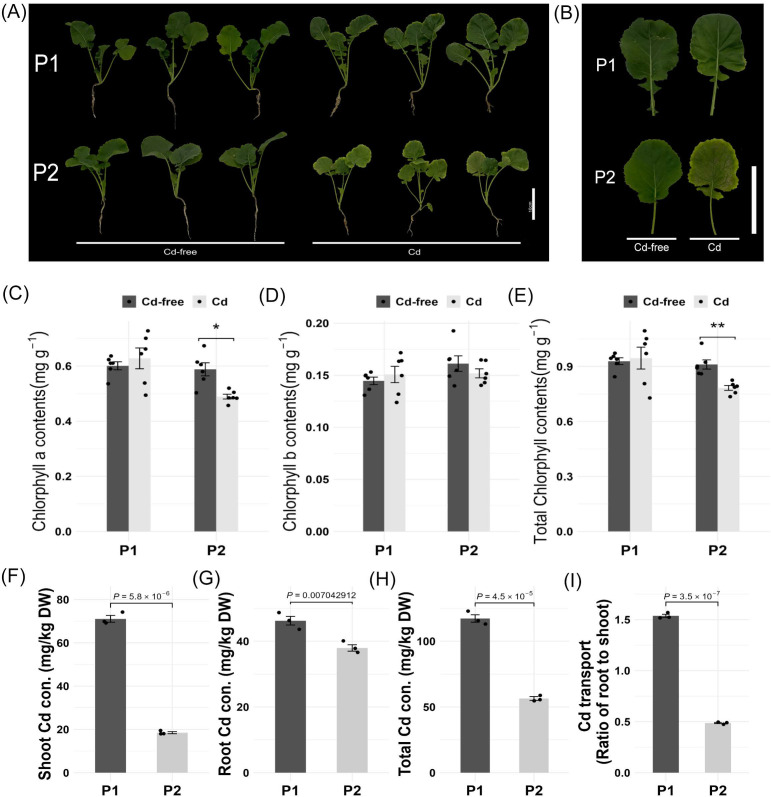
Morphology and Cd (cadmium) accumulation of Cd-enriched (‘P1’) rapeseed line and low-Cd-accumulation (‘P2’) rapeseed line under Cd^2+^ treatment. (**A**) Morphology of the Cd-enriched (‘P1’) rapeseed line and low-Cd-accumulation (‘P2’) rapeseed line under Cd-free (control) and 5 mg/kg Cd^2+^ treatment. Scale bars = 10 cm. (**B**) Images of leaves of the Cd-enriched (‘P1’) and low-Cd-accumulation (‘P2’) rapeseed lines under Cd-free (control) and 5 mg/kg Cd^2+^ treatment. Scale bars = 10 cm. (**C**–**E**) Chlorophyll content of Cd-enriched (‘P1’) and low-Cd-accumulation(‘P2’) rapeseed lines under Cd-free (control) and 5 mg/kg Cd^2+^ treatment. Error bars represent the SE values, * *p* < 0.05, ** *p* < 0.01 unpaired *t*-test; *n* = 6. (**F**–**H**) Cd^2+^ concentrations in the shoot, root, and whole plant of ‘P1’ and ‘P2’. (**I**). Cd transport ratio from root to shoot in ‘P1’ and ‘P2’. Error bars represent the SE values; *p* values for statistical significance are shown as determined by paired *t*-test; *n* = 3.

**Figure 2 ijms-24-10163-f002:**
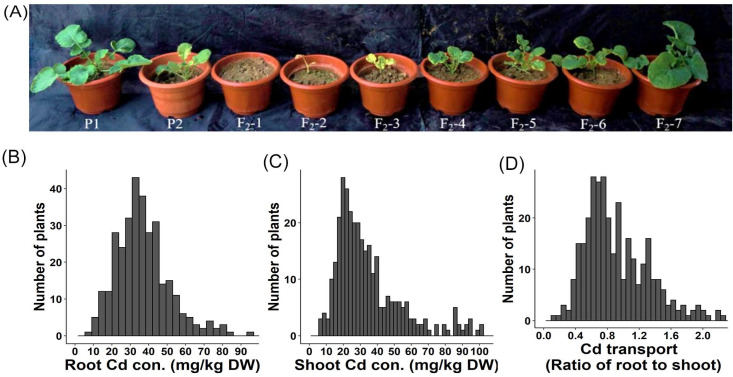
Growth performance and Cd accumulation of F_2_ individuals under Cd^2+^ treatment. (**A**) Growth performance of P1, P2, and F_2_ individuals under 5 mg/kg Cd^2+^ treatment. (**B**) Frequency distribution of Cd^2+^ concentrations in shoots of F_2_ individuals. (**C**) Frequency distribution of Cd^2+^ concentrations in roots of F_2_ individuals. (**D**) Frequency distribution of the Cd^2+^ transport ratio of F_2_ individuals.

**Figure 3 ijms-24-10163-f003:**
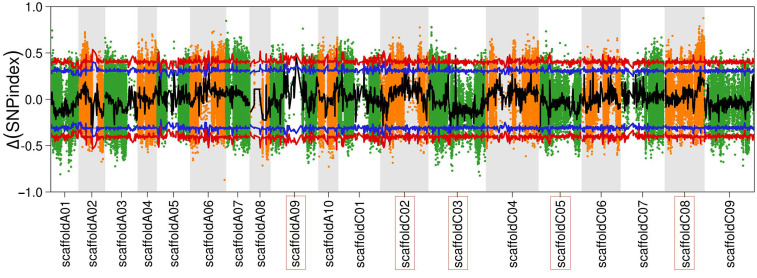
Distribution of genome-wide single nucleotide polymorphisms between the Cd-enriched and low-Cd-accumulation rapeseed lines. The *x*-axis represents the rapeseed chromosome sizes (Mb) while the *y*-axis represents the delta SNP index. The red line is the 99% confidence interval, and the blue line is the 95% confidence interval. The five chromosomes containing significant confidence intervals are marked in red boxes.

**Figure 4 ijms-24-10163-f004:**
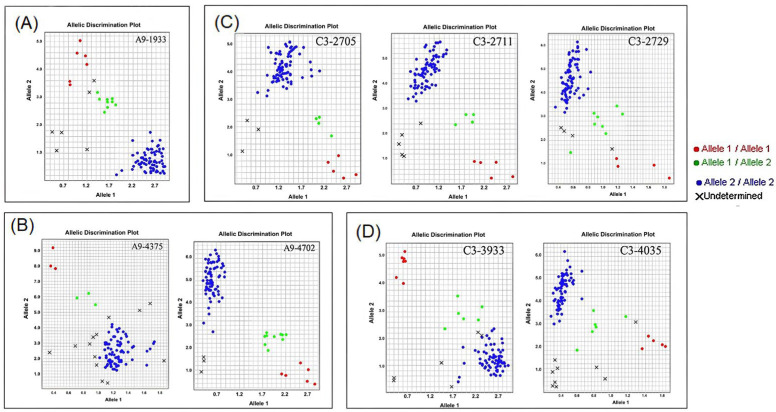
Genotyping results of 89 low-cadmium-accumulation F_2_ individuals on chromosome C03 and chromosome A09. The red dots are cadmium-enriched genotypes, the green dots are heterozygotes, and the blue dots are the low-cadmium-accumulation genotypes; × is the result of invalid genotyping. (**A**) The KASP detection results of QTL Cd-en-qtl1 on chromosome A09. (**B**) The KASP detection results of QTL Cd-en-qtl2 on chromosome A09. (**C**) The KASP detection results of QTL Cd-en-qtl6 on chromosome C03. (**D**) The KASP detection results of QTL Cd-en-qtl7 on chromosome C03.

**Figure 5 ijms-24-10163-f005:**
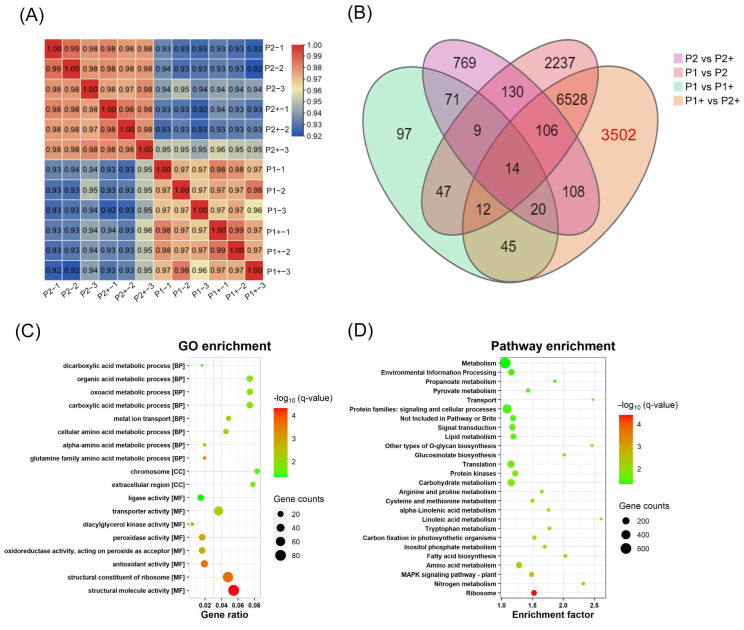
Overview of RNA-sequencing data for the Cd-enriched and low-Cd-accumulation lines. (**A**) Correlation analysis of the RNA-seq samples. ‘P2+’ and ‘P2’ represent the low-Cd-accumulation line with and without 5 mg/kg Cd^2+^ treatment, respectively; ‘P1+’ and ‘P1’ represents the Cd-enriched rapeseed line with and without 5 mg/kg Cd^2+^ treatment, respectively. (**B**) Venn diagrams for DEGs in ‘P1+’ compared with ‘P1’, ‘P2’, and ‘P2+’. (**C**) GO term enrichment analysis of 3502 DEGs. (**D**) KEGG pathway enrichment analysis of 3502 DEGs.

**Table 1 ijms-24-10163-t001:** Significant candidate interval of Δ(SNP index).

QTL Name	Chromosome	Position (bp)	Peak
Cd-en-qtl1	scaffoldA09	18,300,001–19,100,000	0.370743
Cd-en-qtl2	scaffoldA09	35,100,001–35,600,000	0.451405
Cd-en-qtl3	scaffoldC02	32,350,001–32,850,000	0.324665
Cd-en-qtl4	scaffoldC02	34,950,001–35,450,000	0.286806
Cd-en-qtl5	scaffoldC02	44,450,001–44,950,000	0.338758
Cd-en-qtl6	scaffoldC03	30,250,001–30,750,000	−0.35796
Cd-en-qtl7	scaffoldC03	37,050,001–37,550,000	0.472862
Cd-en-qtl8	scaffoldC05	600,001–1,850,000	−0.34606
Cd-en-qtl9	scaffoldC08	47,900,001–48,700,000	0.349622

**Table 2 ijms-24-10163-t002:** List of candidate genes related to Cd accumulation in rapeseed.

Gene ID	SNP or InDel Location	Description
BnaA09G0244400ZS	coding region	Zinc finger A20 and AN1 domain-containing protein
BnaA09G0256600ZS	intergenic region	Polyadenylate-binding protein-interacting protein 12
BnaA09G0263300ZS	coding region	Glutathione S-transferase T3
BnaA09G0264000ZS	coding region	-
BnaC02G0319600ZS	coding region	Putative U5 small nuclear ribonucleoprotein
BnaC02G0319800ZS	coding region	2-dehydro-3-deoxyphosphooctonate aldolase 1
BnaC02G0320100ZS	coding region	Phosphoglycerate kinase
BnaC02G0354400ZS	intergenic region	Putative lipid-binding protein AIR1
BnaC03G0445900ZS	coding region	Photosystem II 5 kDa protein
BnaC03G0447700ZS	coding region	26S proteasome non-ATPase regulatory subunit
BnaC03G0448400ZS	coding region	Receptor-like protein kinase
BnaC03G0448500ZS	intergenic region	-
BnaC03G0449200ZS	coding region	1-deoxy-D-xylulose-5-phosphate synthase
BnaC03G0450200ZS	intergenic region	-
BnaC03G0450600ZS	coding region	Chaperone protein DnaJ16
BnaC08G0421600ZS	coding region	Embryonic stem cell-specific 5-hydroxymethylcytosine-binding protein
BnaC08G0422700ZS	coding region	Polyadenylate-binding protein-interacting protein 7
BnaC08G0422900ZS	coding region	3-ketoacyl-CoA synthase 10
BnaC08G0423000ZS	coding region	Protein FATTY ACID EXPORT 7
BnaC08G0423500ZS	coding region	Replication protein A 70 kDa DNA-binding subunit A-like
BnaC08G0423900ZS	coding region	Protein EARLY FLOWERING 3
BnaC08G0424900ZS	coding region	-
BnaC08G0448600ZS	coding region	Stress enhanced protein 2
BnaC08G0456300ZS	coding region	Uncharacterized LOC105629312
BnaC08G0457500ZS	coding region	-
BnaC08G0458500ZS	coding region	Protein NUCLEAR FUSION DEFECTIVE 6
BnaC08G0466100ZS	coding region	Probable WRKY transcription factor 61
BnaC08G0466900ZS	coding region	Protein LURP-one-related 15
BnaC08G0467300ZS	regulatory region	Transcription factor MYB86
BnaC08G0427700ZS	coding region	UDP-glycosyltransferase 71C3
BnaC08G0457900ZS	coding region	PDDEXK-like family of unknown function
BnaC08G0467400ZS	intergenic region	DnaJ (Hsp40) homolog subfamily C member

## Data Availability

Not applicable.

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
