# Peer review of "Mapping and Identifying Candidate Genes Enabling Cadmium Accumulation in Brassica napus Revealed by Combined BSA-Seq and RNA-Seq Analysis"

_ijms, 2023, doi:10.3390/ijms241210163_

Round 1
Reviewer 1 Report
The authors report on a study to genetically characterize a cadmium-sensitive oilseed rape cultivar and another with little or no sensitivity. After growing the two varieties in pots, they treated them with 5 mg/kg cadmium. They then studied the macroscopic morphological effects (leaves) and genes sensitive to cadmium-induced stress. They applied an omics approach to the genome and transcriptome.
The manuscript has potential but remains very descriptive. The authors have identified genes but have not suggested potential applications, e.g. in agronomy.
The authors claim to have selected 186 canola varieties, but the data are not shown. Why?
Why did they use 5 mg/kg cadmium? Why didn't they do a dose-response curve? Also, should 5 mg/kg be understood as 5 mg of cadmium per 1 kg of soil?
The QTL approach is very important and innovative, I suggest discussing this recent manuscript: Mapping of QTLs controlling barley (Hordeum vulgare L.) agronomic traits under normal conditions and drunk and salinity stress at reproductive stage, 10.1016/j.plgene.2022.100375.
Another critical issue is the statistical approach used for the data in Figure 1. Is the t-student really the best approach?
Minor editing of English language required
Author Response
Comments and Suggestions for Authors:The authors report on a study to genetically characterize a cadmium-sensitive oilseed rape cultivar and another with little or no sensitivity. After growing the two varieties in pots, they treated them with 5 mg/kg cadmium. They then studied the macroscopic morphological effects (leaves) and genes sensitive to cadmium-induced stress. They applied an omics approach to the genome and transcriptome.
Reply: Thank you for your insightful comments and suggestions on our manuscript. We appreciate the time you have taken to provide feedback, and we have carefully considered each of your points. Our responses are provided below:
- Comments: The manuscript has potential but remains very descriptive. The authors have identified genes but have not suggested potential applications, e.g. in agronomy.
Reply: Your suggestion is very good. The use of high cadmium-enriched rapeseed is highly valuable in the remediation of cadmium-contaminated farmland, thus an inclusion of this aspect into the ongoing discussion has been made. Line 268-272, as well as 312-317, and in the conclusion Line 501-505.
- Comments: The authors claim to have selected 186 canola varieties, but the data are not shown. Why?
Reply: Indeed, about ten years ago, a total of 186 rapeseed germplasm resources were identified by a series of screenings, including a laboratory germination stress trail, a 2-year pot trial, and contaminated soil field trail. Finally, cadmium-sensitive and cadmium-enriched materials were screened out. However, in this study, only the phenotypic and genotypic properties of these resulting P1 and P2 rapeseed lines, and their progeny, exhibiting the most significant cadmium sensitivity and insensitivity were examined. This has now been clarified in the text by removing the reference to 186 lines in the abstract and adding the methods for generating the parents are also now included in lines 402-406
- Comments: Why did they use 5 mg/kg cadmium? Why didn't they do a dose-response curve? Also, should 5 mg/kg be understood as 5 mg of cadmium per 1 kg of soil?
Reply: We did experiments with different gradients of cadmium concentration (5 mg/kg, 10 mg/kg, 20 mg/kg, 40 mg/kg, and 60 mg/kg) treatment ten years ago, and cadmium-enriched rapeseed parent “P1” can even bloom and seed normally under cadmium treatment conditions of 60mg/kg. Therefore, considering that a F2 genetic population was planted in this experiment, a relatively low soil cadmium concentration was employed. 5 mg/kg can be understood as 5 mg of cadmium per 1 kg of soil, this has been added in the materials and methods (line 406-413).
- Comments: The QTL approach is very important and innovative, I suggest discussing this recent manuscript: Mapping of QTLs controlling barley (Hordeum vulgare L.) agronomic traits under normal conditions and drunk and salinity stress at reproductive stage, 10.1016/j.plgene.2022.100375.
Reply: Thanks for your suggestions, we discussed and cited this article in the discussion. Line 295-297.
- Comments: Another critical issue is the statistical approach used for the data in Figure 1. Is the t-student really the best approach?
Reply: Thanks for the suggestion, we apologize for the negligence in the description of the material method, it is worth noting that while Figures 1C, 1D, and 1E were analyzed utilizing unpaired t-test., Figure 1F, Figure 1G, Figure 1H and Figure 1I underwent analysis through a paired t-test. This has been corrected in each figure and materials and methods.
Comments on the Quality of English Language
Minor editing of English language required
Reply: Finally, thank you again for reviewing our paper, and sincerely thank you for your language revision questions, we will carefully revise to improve the quality and readability of the paper, we asked Dr. Alice Hayward, a native English speaker, to revise and check the language.

Reviewer 2 Report
Authors definitely did hard work with many interesting data. However, many points need to be clarified and improved.
One of the methodological mistake is that author analysed whole seedlings (RNA). However, it is well known that RNA profile is cell type specific. It needs to be mentioned at least in discussion.
Abstracts.
Please, make it more clear. More explanation require for term cadmium-enchriched parent, cadmium transfer ratio etc.
It is not clear from abstracts what is the reason of crossing, how do you select segregation etc.
Abstracts must be clear itself for almost all reader who less or more familiar with the topic in order to get high audience! Try to avoid very specific term and provide only basic relevant information.
Line 27: “mechanisms of Cd tolerance” - from where come this words? There is no any notification about tolerance before.
Line 33: “chemical fertilizers and pesticides” usually do not contain Cd itself.
Lines 38-43: something wrong with this pretty long sentence: twice 277 nickel plants etc. Please, edit carefully.
Please, provide citation for next sentence (Cu/Cb hyper accumulators).
Line 47: Brassica napus must be in italic.
Lines50-53: complicate sentence, it is better to split. What do you mean as different species here? Other Brassica or different sub-species of napus?
Line 57: “Cadmium toxicity often inhibits plant growth” ??? Toxicity can not inhibit. Cadmium inhibit.
Line 59: “These strategies can generally be divided into cadmium uptake, cadmium
transport, and cadmium accumulation [12,20].” - did you mean inhibition of Cd uptake and transport and accumulation in vacuole compartments?
Line 68: “seeds” is redundant. Gene functioned in plants and inhibit uptake in roots.
Line 69: “Different genes also control Cd accumulation in a tissue‐specific way in a single species.” - redundant sentence, indeed.
Line 95: “ 5mg/kg “ - it is not a new information, you mentioned this in previous sentence, redundant!
Figure 1: I would suggest to use black and white layout for the column.
Line 246: not stress, but pollution.
Lines 272- 276: long sentence with confusing two ROS “windows”. ROS under 5 mg/kg Cd have main effect of organ to organ communications and plant nutrition with carbon (carbon deficiency).
It will be nice to mention in discussion.
Line 317: “Transcription factors such as WRKY and MYB play important roles in regulating plant abiotic stress [58‐61],”. It must be dot in the end, not comma. Moreover, WRKY and MYB regulated not stress, but stress response.
It will be nice to measure Cd contents with sub-cellular resolution.
Despite authors found may possible candidate gene responsible for Cd accumulation, some important information still missing and need to be at least discussed.
How authors can exclude the following mechanism: in Cd-hyperaccumulator line Cd can easy transported to vacuoles (lytic vacuole in the leaf) and thus minimized effect of the Cd on leaf damage.
While in the sensitive cultivar Cd remain in cytoplasm and plastids preventing electron transport in stroma/thylakoid and finally led to carbon deficiency?
Lines 339- 351. There are mistakes here. It is look like copy-paster from 340- 343 to 347-350. However, no information about soil composition, light and temperature conditions etc were present on lines 340-343.
Line 353: “in rapeseed plants” - have you studied another plant species? If no, this is redundant!
some corrections are required.
Author Response
Comments and Suggestions for Authors
Authors did hard work with many interesting data. However, many points need to be clarified and improved.
- Comments: One of the methodological mistake is that author analysed whole seedlings (RNA). However, it is well known that RNA profile is cell type specific. It needs to be mentioned at least in discussion.
Reply: Indeed, the results would be more clear if the roots, stems, leaves and other tissues of rapeseed were sampled separately for RNA sequencing. We apologize for the lack of consideration of our experiments, and we have added a discussion about this part. Line 330-335.
Abstracts.
2.Comments: Please, make it more clear. More explanation require for term cadmium-enriched parent, cadmium transfer ratio etc. It is not clear from abstracts what is the reason of crossing, how do you select segregation etc. Abstracts must be clear itself for almost all reader who less or more familiar with the topic in order to get high audience! Try to avoid very specific term and provide only basic relevant information.
Reply: Thank you very much for your advice, we have rewritten our abstract, including explanation of cadmium enrichment and more clarity on the methods as requested. This will make the structure of the paper clearer and more straightforward.
- Comments: Line 27: “mechanisms of Cd tolerance” - from where come this words? There is no any notification about tolerance before.
Reply: Thank you for your comments, we have rewritten this sentence. Line 28
- Comments: Line 33: “chemical fertilizers and pesticides” usually do not contain Cd itself.
Reply: Generally speaking, natural sources and man-made sources are two sources of soil cadmium pollution, and a variety of anthropogenic sources of soil cadmium pollution, mainly including industrial mining, transportation, manufacture and combustion of cadmium-containing products, irrigation of cadmium-containing sewage in agriculture, and application of cadmium-containing fertilizers (sludge), your opinion is very good, we have modified this sentence to “cadmium-containing chemical fertilizers and products” to make it more accurate. Line 41
- Comments: Lines 38-43: something wrong with this pretty long sentence: twice 277 nickel plants etc. Please, edit carefully.
Reply: Thanks for the suggestion, we have rewritten this sentence to read “Globally, researchers have identified approximately 450 plant species that are capable of accumulating heavy metals, including 277 species that are particularly rich in nickel. Examples of such nickel-rich plants include Cheiranthus and Thlaspi arvense in the Cruciferous family, Chinese leaf subflora in the Euphorbiaceae family, and Senecio in the Asteraceae family. Examples of Copper-rich plants include those found in the Labiatae family, and cobalt-rich plants can be found in the Scrophulariaceae, Myrtaceae and Brassicaceae families amongst others. Other heavy metal-rich plants, such as arsenic, zinc, cadmium, selenium, and manganese, have also been identified” (lines 51-56).
- Comments: Please, provide citation for next sentence (Cu/Cb hyper accumulators).
Reply: Thanks for the suggestion, we have added references to Cu/Cb hyper accumulators. Line 55-58
- Comments: Line 47: Brassica napus must be in italic.
Reply: Thanks for the suggestion, we have made the revision to “Brassica napus L”. Line 59
- Comments: Lines50-53: complicate sentence, it is better to split. What do you mean as different species here? Other Brassica or different sub-species of napus?
Reply: Thanks for the suggestion, we have made the revision to “Different Brassica species have different tolerance to Cd, and vary in the organs that accumulate Cd. The highest distribution of cadmium in rapeseed organs has been found in leaves, followed by stems, roots, and pods, with the lowest accumulation in seeds ”. Line 62-65
- Comments: Line 57: “Cadmium toxicity often inhibits plant growth” ??? Toxicity can not inhibit. Cadmium inhibit.
Reply: Thanks for the suggestion, we have made the revision to “Cadmium often inhibits plant growth”. Line 70
- Comments: Line 59: “These strategies can generally be divided into cadmium uptake, cadmium transport, and cadmium accumulation [12,20].” - did you mean inhibition of Cd uptake and transport and accumulation in vacuole compartments?
Reply: Thanks for the suggestion, It is evident that our expression lacks clarity. We have revised this sentence to enhance its clarity. It now reads “These strategies can generally be categorized into three main types: inhibition of cadmium uptake, inhibition of cadmium transport, and inhibition of cadmium accumulation in vacuole compartments”. Line 72-74
- Comments: Line 68: “seeds” is redundant. Gene functioned in plants and inhibit uptake in roots.
Reply: Thanks for the suggestion, we have made the revision to “whereas in soybean, a different gene, Cda1, is responsible for the uptake of cadmium from the soil through the roots of the plant”. Line 82-83
- Comments: Line 69: “Different genes also control Cd accumulation in a tissue‐specific way in a single species.” - redundant sentence, indeed.
Reply: Thanks for the suggestion, we have deleted this sentence.
- Comments: Line 95: “ 5mg/kg “ - it is not a new information, you mentioned this in previous sentence, redundant!
Reply: Thanks for the suggestion, we have made the revision.
- Comments: Figure 1: I would suggest to use black and white layout for the column.
Reply: Thanks for the suggestion, we have redrawn this figure.
- Comments: Line 246: not stress, but pollution.
Reply: Thanks for the suggestion, we have made the revision.
- Comments: Lines 272- 276: long sentence with confusing two ROS “windows”. ROS under 5 mg/kg Cd have main effect of organ to organ communications and plant nutrition with carbon (carbon deficiency). It will be nice to mention in discussion.
Reply: Indeed, toxicity of Cd has been revealed to lower mineral uptake and photosynthesis in plants, leading to a decline in crop production and quality. We have added this in discussion. Line 282-285.
- Comments: Line 317: “Transcription factors such as WRKY and MYB play important roles in regulating plant abiotic stress [58‐61],”. It must be dot in the end, not comma. Moreover, WRKY and MYB regulated not stress, but stress response.
Reply: Thanks for the suggestion, we have made the revision.
- Comments: It will be nice to measure Cd contents with sub-cellular resolution.
Reply: Indeed, this will help us understand the state of cadmium accumulation in plant cells, which we will use in future research experiments.
- Comments: Despite authors found may possible candidate gene responsible for Cd accumulation, some important information still missing and need to be at least discussed. How authors can exclude the following mechanism: in Cd-hyperaccumulator line Cd can easy transported to vacuoles (lytic vacuole in the leaf) and thus minimized effect of the Cd on leaf damage. While in the sensitive cultivar Cd remain in cytoplasm and plastids preventing electron transport in stroma/thylakoid and finally led to carbon deficiency?
Reply: Your comments are highly valued; we have added a discussion about the possibility of transportation and sequestration of cadmium away from sensitive cellular components in the cadmium tolerant lines, and possible role of the unknown genes in this or other mechanisms. Lines 278-279 and 376-379
- Comments: Lines 339- 351. There are mistakes here. It is look like copy-paster from 340- 343 to 347-350. However, no information about soil composition, light and temperature conditions etc were present on lines 340-343.
Reply: Thanks for the suggestion, we apologize for the oversight, we have rewritten the plant planting conditions. Line 363-369
- Comments: Line 353: “in rapeseed plants” - have you studied another plant species? If no, this is redundant!
Reply: Thanks for the suggestion, we have made the revision.
Comments on the Quality of English Language: some corrections are required.
Reply: again, thank you for reviewing our paper, and sincerely thank you for your language revision questions, we will carefully revise to improve the quality and readability of the paper.

Round 2
Reviewer 1 Report
The authors have significantly improved the manuscript following the directions and suggestions of the Reviewers. The manuscript can be published.
Author Response
Thank you again for your insightful comments and suggestions on our manuscript.
Reviewer 2 Report
Line 49: accumulated nickel, not rich.
Line 73: sequestration in vacuole, NOT inhibition!
Line 80: "Different species of plants" ? Different plant species.
Fiigure 1, C, D - layout!
Line 280: grammar.
Minor edition during proof-reading.
Author Response
Comments and Suggestions for Authors
Comments 1:Line 49: accumulated nickel, not rich.
Reply: Thanks for the suggestion, we have made the revision to “nickel-accumulated plants”. Line 49
Comments 2:Line 73: sequestration in vacuole, NOT inhibition!
Reply: Thanks for the suggestion, we have made the revision to “and sequestration of cadmium in vacuole compartments”. Line 74-75
Comments 3:Line 80: "Different species of plants" ? Different plant species.
Reply: Thanks for the suggestion, we have made the revision to “Different plant species behave somewhat differently under cadmium stress”. Line 81
Comments 4:Fiigure 1, C, D - layout!
Reply: Thanks for the suggestion, we have readjusted the layout of Figure 1.
Comments 5:Line 280: grammar.
Reply: Thanks for the suggestion, we have made the revision to “Studies have shown that in Cd-hyperaccumulator plants, Cd is often transported to vacuoles, particularly the lytic vacuole located in the leaf. This process helps to minimize the damaging effects on the leaf caused by exposure to Cd.” Line 279-281
Comments on the Quality of English Language
Minor edition during proof-reading.
Reply: Sincerely thank you for your language revision comments, we have carefully revised to improve the quality and readability of the paper.
